# The phylodynamics of SARS-CoV-2 during 2020 in Finland

Phuoc Truong Nguyen [1,11], Ravi Kant [1,2,11], Frederik Van den Broeck [3,4,11], Maija T. Suvanto [1,2], Hussein Alburkat [1], Jenni Virtanen [1,2], Ella Ahvenainen [1], Robert Castren[1], Samuel L. Hong [3], Guy Baele [3], Maarit J. Ahava [5], Hanna Jarva[5,6,7], Suvi Tuulia Jokiranta [6,7], Hannimari Kallio-Kokko[5], Eliisa Kekäläinen [5,6], Vesa Kirjavainen[5], Elisa Kortela[8], Satu Kurkela[5], Maija Lappalainen[5], Hanna Liimatainen[5], Marc A. Suchard [9], Sari Hannula[10], Pekka Ellonen[10], Tarja Sironen [1,2], Philippe Lemey [3✉], Olli Vapalahti [1,2,5✉] & Teemu Smura [1,5✉]

## Abstract

**Background** Severe acute respiratory syndrome coronavirus 2 (SARS-CoV-2) has caused millions of infections and fatalities globally since its emergence in late 2019. The virus was first detected in Finland in January 2020, after which it rapidly spread among the populace in spring. However, compared to other European nations, Finland has had a low incidence of SARS-CoV-2. To gain insight into the origins and turnover of SARS-CoV-2 lineages circulating in Finland in 2020, we investigated the phylogeographic and -dynamic history of the virus.

**Methods** The origins of SARS-CoV-2 introductions were inferred via Travel-aware Bayesian time-measured phylogeographic analyses. Sequences for the analyses included virus genomes belonging to the B.1 lineage and with the D614G mutation from countries of likely origin, which were determined utilizing Google mobility data. We collected all available sequences from spring and fall peaks to study lineage dynamics.

**Results** We observed rapid turnover among Finnish lineages during this period. Clade 20C became the most prevalent among sequenced cases and was replaced by other strains in fall 2020. Bayesian phylogeographic reconstructions suggested 42 independent introductions into Finland during spring 2020, mainly from Italy, Austria, and Spain.

**Conclusions** A single introduction from Spain might have seeded one-third of cases in Finland during spring in 2020. The investigations of the original introductions of SARS-CoV-2 to Finland during the early stages of the pandemic and of the subsequent lineage dynamics could be utilized to assess the role of transboundary movements and the effects of early intervention and public health measures.

### Plain language summary

The severe acute respiratory syndrome coronavirus 2 (SARS-CoV-2) has caused millions of infections and deaths since it began spreading globally in late 2019. Unlike other European countries, during early pandemics Finland had relatively few coronavirus cases. We investigated how and from where SARS-CoV-2 arrived in Finland in early 2020. Viruses mutate over time and SARS-CoV-2 viruses with different mutations are described as variants. We assessed the proportions of different SARS-CoV-2 variants over time by studying the different mutations occurring combined with travel history data. We found that the first epidemic wave was seeded by 42 viral introductions (mainly from Spain, Italy and Austria), including one that caused a third of all COVID-19 infections. Our results show that international travel was a major contributor to the spread of SARS-CoV-2 in Finland.

[1] Department of Virology, Faculty of Medicine, University of Helsinki, Helsinki, Finland. [2] Department of Veterinary Biosciences, Faculty of Veterinary Medicine, University of Helsinki, Helsinki, Finland. [3] Department of Microbiology, Immunology and Transplantation, Rega Institute, KU Leuven, Leuven, Belgium. [4] Department of Biomedical Sciences, Institute of Tropical Medicine, Antwerp, Belgium. [5] HUS Diagnostic Center, HUSLAB, Clinical Microbiology, University of Helsinki and Helsinki University Hospital, Helsinki, Finland. [6] Translational Immunology Research Program, University of Helsinki, Helsinki, Finland. [7] Department of Bacteriology and Immunology, University of Helsinki, Helsinki, Finland. [8] Infectious Diseases, University of Helsinki and Helsinki University Hospital, Helsinki, Finland. [9] Departments of Biomathematics, Biostatistics and Human Genetics, University of California, Los Angeles (UCLA), Los Angeles, CA, USA. [10] Institute for Molecular Medicine Finland (FIMM), Helsinki, Finland. [11]These authors contributed equally: Phuoc Truong Nguyen, Ravi Kant, Frederik Van den Broeck. ✉email: philippe.lemey@kuleuven.be; olli.vapalahti@helsinki.fi; teemu.smura@helsinki.fi

Severe acute respiratory syndrome coronavirus 2 (SARS-CoV-2) belonging to betacoronaviruses (genus *Betacoronavirus*) causes coronavirus disease 2019 (COVID-19), a respiratory infection with severe cases leading to respiratory failure and multiorgan manifestations in humans, and is responsible for the current socially and economically devastating pandemic. The virus has infected more than 184 million people in 221 countries and has caused over 3.9 million deaths as of July 5, 2021[1]. The virus is similar to other betacoronaviruses in terms of a relatively high evolutionary rate ($\sim 9.8 \times 10^{-4}$ substitutions per site per year)[2] leading to the emergence of multiple viral lineages circulating the globe. Viral lineages may become more common in a given host population due to selective advantages or by chance e.g., due to the founder effect or genetic drift. Despite there being currently a plethora of viral lineages, only a small proportion of these are classified as variants of concern (VOCs), i.e. are considered to have enhanced transmissibility, pathogenicity, evasion of immune responses, or resistance to vaccines[3]. Currently, this category contains only the lineage B.1.617.2 (Delta)[4] first detected in India[5] and B.1.1.529 (Omicron) discovered in Botswana[6], but has previously included the lineages B.1.1.7 (Alpha) first detected in the United Kingdom (UK)[7], B.1.351 (Beta) first detected in South Africa[8], and P.1 (Gamma) first detected in Brazil[9].

The first Finnish SARS-CoV-2 case was detected on January 29, 2020, from a tourist from Wuhan, China[10] (Fig. 1), however, this infection did not lead to onward transmissions. The first epidemic wave in Finland began in week 9 (end of February 2020), peaked during week 14 (beginning of May) and ended by week 24 (early June). The incidence was low during the following summer (from mid-June to July). The second epidemic wave began in week 32 (beginning of August 2020) and lasted approx. until week 45 (early November). For additional information about the introduction and spread of SARS-CoV-2 in Finland during 2020, see Supplementary Note 1.

In order to gain insight into the geographic source and relative contribution of viral introductions that seeded the first wave epidemic in Finland as well as study phylodynamic aspects, such as the genetic diversity and lineage turnover, of circulating viruses during 2020, we sequenced 1,597 SARS-CoV-2 genomes from Finland and analyzed these using travel-aware Bayesian phylogeographic approaches including 1643 genomes from 17 European countries.

## Methods

### Sequencing and analyses of Finnish SARS-CoV-2 genomes.
Research data for this report consists of SARS-CoV-2 genomes ($n = 1,597$) that were sequenced from SARS-CoV-2 PCR positive patient samples diagnosed in HUS Diagnostic Center, HUSLAB, University of Helsinki and Helsinki University Hospital (Fig. 2). This study was approved by the Research Administration of Helsinki University Hospital (HUS/32/2018 and HUS/157/2020) and no identifiable patient data were described in this study. As this was a retrospective registry study with no patient intervention, ethics committee approval and informed consent were not required by Finnish national legislation in accordance with the Medical Research Act of Finland 488/1999. RNA was reverse-transcribed to cDNA with the LunaScript RT SuperMix kit (New England Biolabs). Primer pools[11] targeting SARS-CoV-2 were designed using the PrimalScheme tool[12] (Supplementary Data 1) and PCR was conducted with PhusionFlash PCR master mix (Thermo Scientific). Sequencing libraries were prepared with NEBNext ultra II FS DNA library kit (New England Biolabs) according to the manufacturer's instructions and sequenced with Illumina NovaSeq and MiSeq. Due to HUSLAB initially being the only clinical laboratory sequencing patient samples, some of the virus sequences originate from outside the HUS area e.g., from testing points on the border. The collection period was from spring to fall 2020 and the sampling was random. However, the data might be biased for the most severe cases of SARS-CoV-2, and there was no contact tracing for our data at that time. Consensus sequence data for Finnish SARS-CoV-2 was computed and classified either with the HAVoC pipeline[13] (which utilizes fastp[14] for quality filtering, BWA-MEM[15] for assembly, LoFreq[16] for variant calling and SAMtools[17] for consensus calling) or a modified pipeline consisting of Jovian[18] and pangolin[19]. Sequences were then submitted to the GISAID database. Clade and lineage assignment was done using Nextclade[20] and pangolin.

For the phylogenetic analysis of Finnish fall sequences, in addition to the local sequence data between weeks 32–38 ($n = 77$), a global reference dataset of SARS-CoV-2 genomes ($n = 745$) was selected from sequences from other countries ($n = 20,720$) from the same time period. These were obtained from the GISAID database (Supplementary Data 2). Viral sequences from fall of 2020 were aligned with MAFFT[21] and the phylogenetic tree was computed with a SARS-CoV-2 version IQ-TREE (version 2.1.3)[22] with 1,000 bootstraps and with the most optimal substitution model using ModelFinder[23]. Finally, the tree was visualized in R with ggtree[24] and ggtreeExtra[25].

### Bayesian time-measured phylogeographic analyses.
In order to infer the geographic source(s) of SARS-CoV-2 lineages contributing to the first wave in Finland, we extended our dataset of Finnish genomes with genomes available for other European countries (Fig. 2). A recent phylogeographic analysis demonstrated that SARS-CoV-2 spread in Europe was strongly predicted by Google mobility flows[26]. To inform our sampling, we therefore turned to the Google COVID-19 Aggregated Mobility Research Dataset containing anonymized mobility flows aggregated over users who have turned on the Location History setting (on a range of platforms[27]). Aggregated mobility flows between Finland and all other European countries were summarized between January and April 2020, and we selected the following 16 countries that were responsible for 95% of international travels from and to Finland: Estonia, Latvia, Norway, Hungary, Poland, Turkey, Sweden, Netherlands, Austria, Denmark, Italy, Germany, Switzerland, Spain, France and the United Kingdom. For these countries, we downloaded the available SARS-CoV-2 genomes from GISAID on April 17, 2020. For six countries (Estonia, Latvia, Norway, Hungary, Poland and Turkey) represented by a relatively small number of genomes, we decided to augment our dataset with genomes from GISAID with a sampling date up to April 31, 2020.

We selected only sequences from the B.1 lineage with the D614G mutation for the analyses. We removed duplicate genomes for each country using SeqKit (version 0.11)[28]. For Finland, we retained duplicate genomes when these were sampled from cases with different travel histories. All genomes were aligned using MAFFT[21] and trimmed at the 5′ and 3′ ends. We then subsampled each country proportionally to the cumulative number of cases on April 17, 2020 by setting an arbitrary threshold of 7.5 genomes per 10,000 cases, with a minimum number of 100 sequences per country. For the 6 countries where the number of unique genomes was below 100, all genomes were included in the analysis. To maximize the spatial and temporal coverage of the subsampling, we partitioned each country's genome pool by week and sampled as evenly as possible, selecting sequences from a different region within the country when available. We checked the resulting dataset for potential outliers with a root-to-tip regression using TempEst (version 1.5.3)[29] on a

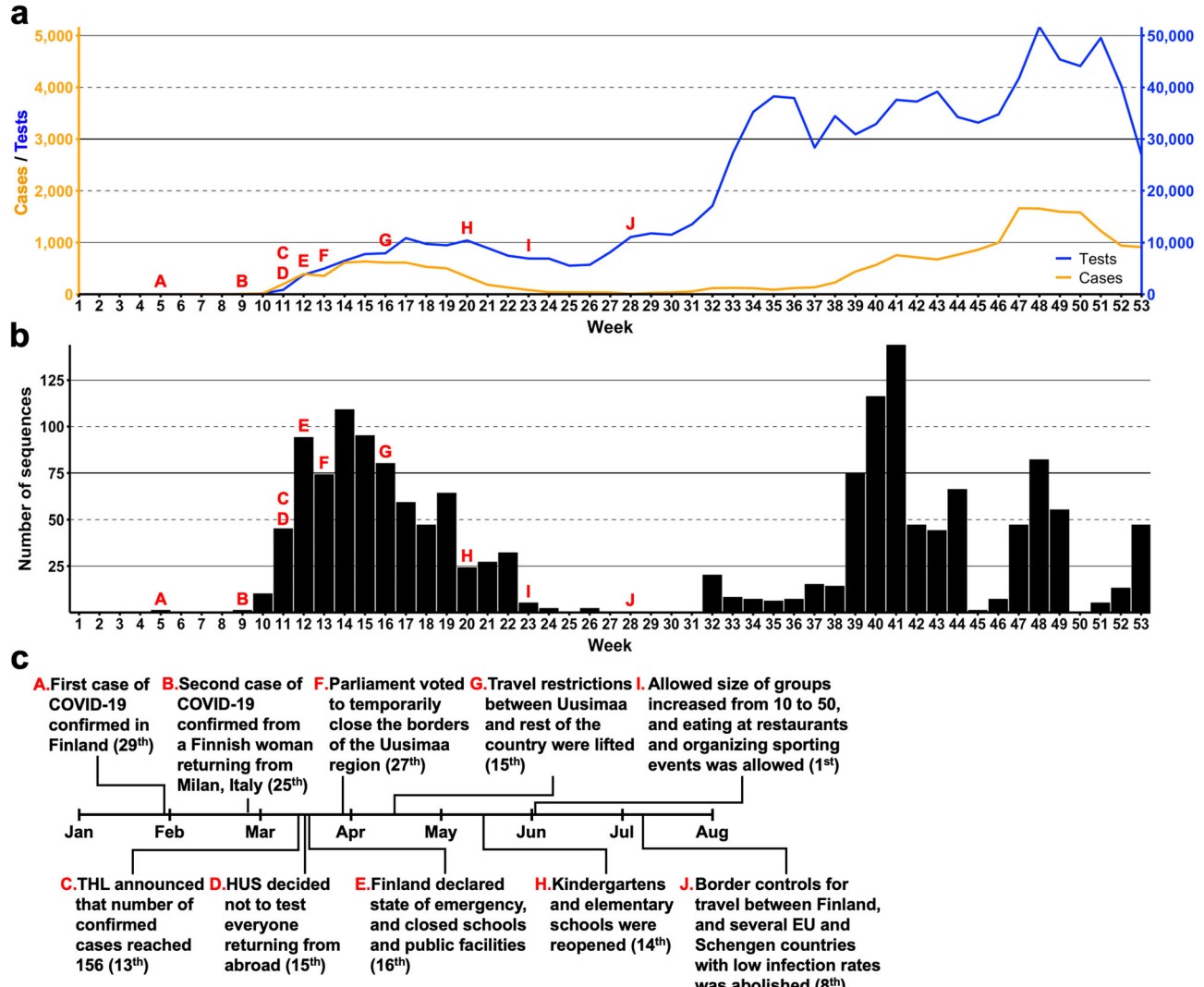

**Fig. 1 Weekly SARS-CoV-2 statistics and general timeline of Finland in 2020.** The number of PCR tests (total $n = 967,885$) and positive findings (total $n = 21,731$) based on the COVID-19 infectious diseases registry of the Finnish Institute for Health and Welfare (THL) are shown in panel (**a**). The color of lines matches their respective axes, i.e. the axis indicating number of tests is on the right and number of positive cases on the left. The number of SARS-CoV-2 sequences submitted to GISAID (total $n = 1,597$) are displayed in panel (**b**). Panel (**c**) depicts the general timeline of the arrival of SARS-CoV-2 in Finland and the subsequent responses by the Finnish government and health authorities, which are indicated by letters A-J in panels (**a**) and (**b**). Exact dates for each response are mentioned within brackets. This information is based on public records by THL. HUS = Hospital District of Helsinki and Uusimaa.

maximum likelihood inferred using IQ-TREE (version 2.0.3)[22], and removed 9 genomes. The final dataset consisted of 1,643 genomes out of an initial 8,513 genomes in spring only. Total, unique and downsampled number of genomes by country are given in Supplementary Table 1. All genomes were associated with exact sampling dates, except for the four genomes from Estonia that were sampled in March 2020.

We performed Bayesian evolutionary reconstruction of timed phylogeographic history using BEAST (version 1.10)[30] incorporating genome sequences, their country and date of sampling, Google mobility data, and individual travel history[31,32]. Uncertainty in the sampling time for the four Estonian genomes was accommodated by sampling uniformly across the reported collection month in the Markov chain Monte Carlo (MCMC) analysis. We modeled sequence evolution using a strict molecular clock model and an HKY nucleotide substitution model[33] with gamma-distributed rate variation among sites[34]. We assumed an exponential growth coalescent model as the tree-generative process prior because we only used viral sequences sampled up

to the 17th of April, which means that the large majority of sequences were sampled from a viral population experiencing exponential growth. To demonstrate that our results are not sensitive to the choice of this coalescent prior, we have also repeated the BEAST phylogeographic reconstruction with travel history using the Skygrid as a tree prior.

Our phylogeographic model incorporated the country of sampling as discrete traits associated with the sampled genomes, and following a recent European SARS-CoV-2 phylogeographic analysis[26], we adopted a generalized linear model (GLM) specification to parametrize each rate of among-location movement as a log linear function of the total Google mobility flows (i.e. relative population flow between each pair of geographical areas over a given time interval) for the January-April, 2020 period. Total mobility flows were log-transformed and standardized after adding a pseudocount to each entry in the matrix. The main goal of our GLM extension was to obtain well-informed phylodynamic estimates. To demonstrate that our GLM parameterization is a better option than the standard inference

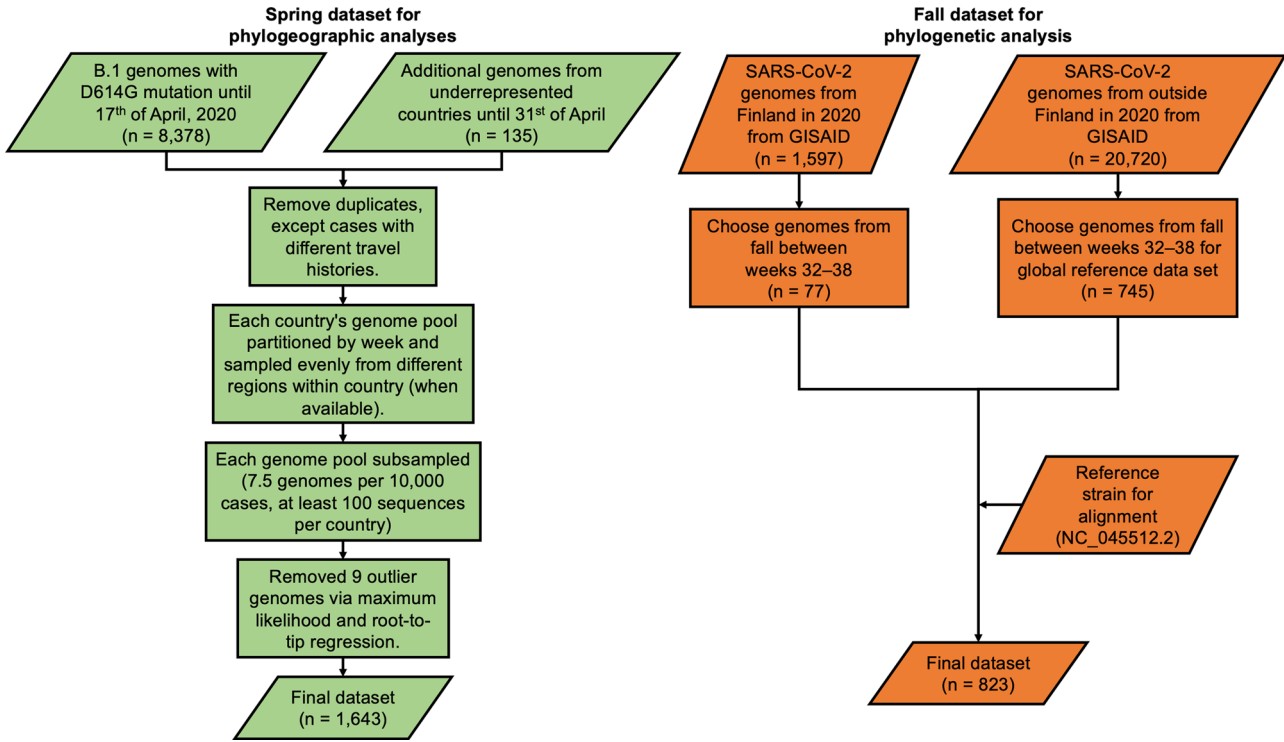

**Fig. 2 Flowchart of sequence data acquisition and sampling and/or selection for spring and fall analyses.** Available Finnish SARS-CoV-2 sequences (Supplementary Data 2) were divided into spring and fall datasets based on the peaks in COVID-19 cases during 2020.

procedure with BSSVS, we have estimated marginal likelihoods using a path sampling (PS) and stepping stone sampling (SS) approach. To make this procedure efficient for the large data set investigated here, we averaged over the same set of empirical trees for both parameterizations. Our results demonstrated that the GLM model (−1956.98 (PS) and −1956.78 (SS) log marginal likelihoods) outperforms the standard model with BSSVS (−2186.41 (PS) and −2186.39 (SS) log marginal likelihoods) by over 200 log marginal likelihood units.

As the ancestral reconstruction of locations depends on the availability of samples, over- or undersampling of sequences from a given location can greatly impact the estimated ancestral locations[31]. To mitigate sampling bias and improve the location-transition history reconstructions, we augmented our elementary phylogeographic model by incorporating travel history information obtained from 44 cases that returned to Finland from Austria ($n = 20$), Italy ($n = 13$), Spain ($n = 7$), Estonia ($n = 1$), Germany ($n = 1$), Switzerland ($n = 1$) and United Kingdom ($n = 1$).

We also investigated how unsampled diversity for six European countries or oversampling of Finnish SARS-CoV2 diversity may impact our phylogeographic reconstructions. Building on our extended phylogeographic model including sampling locations and individual travel histories, we incorporated unsampled taxa for the undersampled countries Estonia ($n = 96$ taxa added), Latvia ($n = 83$), Norway ($n = 56$), Hungary ($n = 54$), Poland ($n = 46$) and Turkey ($n = 41$) to arrive at a minimum of 100 genomes for all countries. Unsampled taxa without observed sequence data were added with associated location and sampling times, for which we randomly sampled dates from case count distributions per country. For this analysis, we also downsampled the Finnish genome dataset to 100 taxa, while ensuring we incorporated the 44 samples with known travel histories.

We performed inference under the full model specification using MCMC sampling while employing the BEAGLE library (version 3)[35] to increase computational performance. Because MCMC burn-in takes considerable computational time due to the

size of our dataset, with the tree topology representing the most challenging parameter for convergence, we start our analyses with a standard BEAST model considering only sequence evolution (strict molecular clock model, HKY nucleotide substitution model, and exponential growth tree prior). The resulting phylogenetic tree was subsequently used as a starting tree in our phylogeographic analyses. Multiple independent MCMC runs were run to ensure that their combined posterior samples achieved effective sample sizes (ESSs) larger than 100 for all continuous parameters. Transition histories were summarized using the tree sample tool, TreeMarkovJumpHistoryAnalyzer, implemented in BEAST to collect Markov jumps[36] and their timings from a posterior tree distribution annotated with Markov jumps histories[26].

**Reporting summary.** Further information on research design is available in the Nature Research Reporting Summary linked to this article.

## Results

**Lineage distribution and turnover during spring and fall epidemic waves.** During the year 2020, there were 37,145 laboratory-confirmed COVID-19 cases in Finland[37], of which 21,730 (58.5%) were diagnosed in the HUS Diagnostic Center[37]. We sequenced a total of 1597 SARS-CoV-2 genomes from the year 2020, which accounts for over 7.4% of all positive samples from the HUS area and represents 4.3% of positive samples from Finland in 2020.

By week 11 (mid-March), all major clades of SARS-CoV-2 had been introduced to Finland. These were the GISAID clades G, GR, L and V, of which the first corresponds with Nextstrain clade 20A, the second with 20B, and the two latter ones with 19 A. All aforementioned GISAID clades belong to the major lineage B based on pangolin classification[38]. By week 16 (mid-April), the lineages that contain the D614G substitution in the spike protein (20A, 20B and 20C or B.1) became dominant

(Supplementary Fig. 1). Out of these, Nextstrain clade 20C grew relatively fast into the dominant clade, starting from week 14 (beginning of April), when the number of 20C detections among the weekly cases nearly tripled from 17 (23%) to 59 (54%) from the previous week (Supplementary Fig. 1). While the number of cases (and sequences) were low during June and July, all of them represented this cluster. The clade was subsequently replaced by lineages 20A and 20B later in fall.

By August 2020, there was sufficient global genetic diversity in SARS-CoV-2 for a more fine-grained analysis using Pango lineage classification (Supplementary Fig. 2). During the spring and fall time, most sequences belonged to lineage B.1 (669 of a total of 1,597, 41.89%). Notably throughout fall (weeks 32–42), lineages B.1.36.22 ($n = 294$, 51.67%) and B.1.463 ($n = 120$, 21.09%), that consist almost exclusively of viruses sequenced from Finland, formed the majority of detected lineages ($n = 569$). The third most common lineage, B.1.160 ($n = 68$, 11.95%), is a large European lineage found in many countries[39,40].

During the initial stages of the second wave of SARS-CoV-2 in fall (weeks 32–38), several lineages were detected approximately in equal proportions e.g., many lineages constituted ca. 14% of cases in week 34, until the number of cases rapidly increased in week 39 (end of September). During this time, three lineages, Finland-specific B.1.36.22 and B.1.463, as well as the Pan-European B.1.160 grew in prominence and became dominant in Finland. A phylogenetic tree constructed using maximum-likelihood inference including Finnish SARS-CoV-2 sequences from the fall of 2020 (Supplementary Fig. 3) show that most of these sequences (52 out of 77, 67.53%) fall into three major clusters with each containing viruses belonging to either lineage B.1.36.22 ($n = 28$), B.1.463 ($n = 12$) or B.1.160 ($n = 12$). These major lineages from Finland form monophyletic clusters each, suggesting a single ancestor for them circulating in Finland during fall 2020. Notably, 20E(EU1) (lineage B.1.177), despite being a widespread clade in Europe during the summer of 2020[40,41], was not detected in high numbers in fall (10 cases in August), suggesting that it did not largely contribute to the rise of cases during the second wave in Finland.

**Travel-aware phylogeographic inference of early COVID-19 spread in Finland**. In order to gain insights into the early stage of COVID-19 spread in Finland, we traced the geographic sources of viral introductions into Finland using travel-aware Bayesian phylogeographic reconstructions[31,32] of dispersal patterns between pairs of 17 European countries during the first epidemic wave (Fig. 3a). Our analysis included 1643 genome sequences from the B.1 lineage, including 333 genome sequences from Finland that were sampled until the 17th of April, 2020 (Supplementary Data 2). Sequences were associated with their country and date of sampling. For 13% of the sampled sequences in Finland (see methods section), we incorporated individual travel history information to augment the phylogeographic model. This was done by introducing ancestral nodes in the phylogeny that are associated with locations visited by travelers, a procedure that mitigates sampling bias and improves the location-transition history reconstructions of SARS-CoV-2[31,32]. To further inform our reconstructions and achieve the best possible resolution, we also adopted a generalized linear model (GLM) parametrization with Google mobility data as the predictor, building on previous insights that Google mobility is the best mobility predictor to inform SARS-CoV-2 phylogeographic reconstructions[41]. We used model testing to confirm that a standard inference procedure including a Bayesian stochastic search variable selection (BSSVS) is inferior to using the GLM parameterization (see methods section). In addition, a BSSVS procedure applied to this predictor

in the GLM, offered support that mobility data was at least superior to a uniform rates model (with an inclusion probability equal to 1 and a posterior mean log coefficient of 0.5, 95% highest posterior density, HPD interval = 0.36–0.65).

Our phylogeographic reconstructions revealed a total of 42 individual introductions into Finland (95% HPD, interval = [36–47]). Of all estimated introductions into Finland, the majority occurred during the second week of March and originated from Italy (12 introductions, 95% HPD interval = [9–16]), Austria (8 introductions, 95% HPD interval = [6–10]) and Spain (8 introductions, 95% HPD interval = [7–10]) (Fig. 3b). Germany, Sweden, Switzerland, France, the United Kingdom and Denmark each accounted for 1–3 introductions (Fig. 3b). The average number of estimated introductions from Turkey, the Netherlands, Latvia, Estonia, Poland, Norway and Hungary was below 1 (ranging between 0.062 for Latvia and 0.653 for Turkey) and their HPD interval included zero (Fig. 3b), offering little support for viral introductions from these countries. The pattern of viral introductions mainly from Italy, Austria and Spain during the first wave largely matched our epidemiological records with travel history data, as 40 out of 44 imported cases returned from these countries (Fig. 3b). Notably, Austria showed at least twice the number of travel history entries ($n = 20$) compared to viral introductions ($n = 8$) as estimated from our Bayesian phylogeographic analyses (Fig. 3b). Close inspection of the phylogenetic tree as obtained from our Bayesian reconstructions revealed that 12 out of 20 cases returning from Austria clustered tightly within two subclades belonging to a larger cluster of predominantly Austrian genomes (Supplementary Fig. 4), suggesting that these Finnish patients may have picked up the infections from the same source.

Our analysis identified 35 independent introduction events out of 42 (81%) (95% HPD interval = [30–39]) resulting in relatively few sampled Finnish descendants (≤10), including 16 singleton introductions (95% HPD interval = [12–20]). Hence, the large majority of introductions account for a relatively small number of the lineages we sample, a pattern typically observed for all European countries (Fig. 4a). This highlights extensive heterogeneity in SARS-CoV-2 transmission dynamics underlying the establishment of local transmission chains. While the largest number of independent introduction events originated from Italy (Fig. 4b), we identified one introduction from Spain that gave rise to 119 (95% HPD interval = [100–134]) unique genomes sampled in Finland (Fig. 4c), indicating that one third of our first wave sample traces back to a viral lineage originating from Spain. We note that the cumulative number of descendants as shown in Fig. 4c does not include duplicate sequences that were removed prior to the phylogeographic analyses. This involves 27, 28 and 15 additional Finnish descendants for introductions from Spain, Italy and Austria, respectively. As duplicate sequences were similarly distributed across the three countries, the overall pattern observed in Fig. 4c is not affected, namely that a Spanish introduction seeded considerably more Finnish descendants compared to Italian and Austrian introductions. Descendant taxa from this single Spanish introduction belong to Nextstrain clade 20A, which is the second most dominant clade during the first wave epidemic in Finland (Supplementary Fig. 1). The most dominant clade during the first epidemic wave in Finland, clade 20C with D936Y spike mutation, clustered together with Swedish sequences. However, the posterior probability for this clade was low and, therefore, the origin of this clade remained unresolved.

Our phylogeographic reconstructions based on 1,643 sampled genomes may potentially suffer from the impact of sampling bias. For instance, while one of the Finnish COVID-19 cases returned from the neighbouring country of Estonia, we inferred no direct viral movements to Finland from Estonia (Fig. 3b), a country that is severely underrepresented by viral genomes ($n = 4$) compared

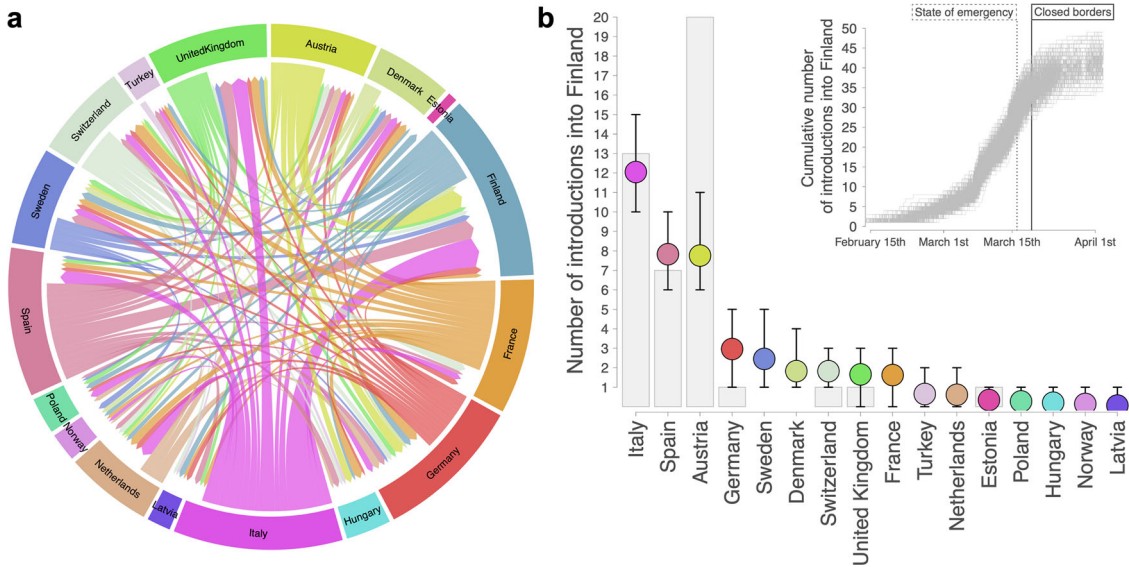

**Fig. 3 Discrete phylogeographic reconstruction of SARS-CoV-2 introductions into Finland during the first wave epidemic. a** Circular migration flow plot based on the posterior expectations of the Markov jumps between 17 country-level locations, including Finland and 16 selected European countries. Migration flow out of a particular location starts close to the outer ring and ends with an arrowhead more distant from the destination location. **b** Mean and 95% highest posterior density (HPD) number of transitions to Finland from each of the 16 selected European countries, as estimated from 1,000 trees subsampled from the posterior distribution. Gray bars indicate the number of cases with travel history data returning from each country. Inset shows the cumulative number of introductions into Finland summarized from a posterior sample of phylogeographic trees. Dashed line indicates the day (16th of March) when the Finnish authorities declared a state of emergency due to COVID-19. Full line indicates the day (19th of March) when the Finnish authorities announced a restriction of passenger traffic at Finland's borders.

to most other countries. To explore the sensitivity of our phylogeographic reconstructions to sampling bias, we incorporated unsampled taxa for 6 locations (Estonia, Latvia, Norway, Hungary, Poland and Turkey) that were represented by less than 60 sequences (other European countries were represented by at least 100 sequences) (Supplementary Table 1), resulting in a dataset of 2019 taxa. Tip ages ("sampling times") were specified as randomly sampled dates from the case count distributions per undersampled country. In addition, as Finland is severely oversampled according to case counts in spring (13.87% versus 0.13–5.79% for the 16 selected European countries) (Supplementary Table 1), we also performed an analysis including the unsampled taxa and in which Finnish genomes were downsampled from 333 to 100 sampled genomes, for a total of 1,786 genomes.

Results obtained from reconstructions without unsampled taxa (Fig. 4), with unsampled taxa (Supplementary Fig. 5A) and with unsampled taxa and downsampled Finnish taxa (Supplementary Fig. 5B) were largely similar in terms of the total number of introductions and the dominant contribution of viral introductions from Italy, Austria, and Spain. However, in contrast to the reconstructions with the full set of Finnish taxa (Fig. 1b and Supplementary Fig. 5A), the reconstruction with downsampled Finnish taxa (Supplementary Fig. 5B) offered support for additional viral introductions from both Estonia (2 introductions, 95% HPD interval = [0–3]) and Latvia (2 introductions, 95% HPD interval = [0–3]). The largely similar results from reconstructions with and without unsampled taxa suggests limited impact of sampling bias on our Bayesian phylogeographic reconstructions.

Finally, we investigated the sensitivity of our timed phylogeographic reconstructions to the choice of an exponential growth coalescent tree prior. For this purpose, we compared travel-aware phylogeographic reconstructions without unsampled taxa using both the Skygrid model[42] and exponential growth model. This comparison shows that the number and timing of

introductions is similar under both coalescent models (Supplementary Figs. 6 and 7).

## Discussion

Finland has had a low incidence of SARS-CoV-2 cases compared to most European countries, including the neighboring countries Sweden and Russia. Intriguingly, during both spring and fall epidemic waves, the majority of infections were caused by a limited number of viral lineages. These included a subcluster of clade 20C with spike protein D936Y substitution during spring, and the predominance of lineages B.1.36.22, B.1.463, and B.1.160 during fall. The prevalence of D936Y increased concurrently in Sweden[43], and the mutation has also been detected in Wales (as early as March 15, 2020) in association with the D614G mutation, as well as in England, and with low frequencies in Denmark, Poland and the United States[44]. The emergence and rapid spread of this mutation might be caused by periodic positive selection pressures[43] despite its destabilizing effect on post-fusion spike protein assembly due to a loss of a salt bridge between monomers[44]. The dominant fall lineages in turn have been detected only sporadically in other countries (B.1.36.22 in Norway, Denmark, Latvia and Canada, and B.1.463 in Denmark)[39]. These suggest that despite multiple introductions of the virus to an immunologically naive population, only few of these resulted in long transmission chains. This is consistent with the well-known super-spreading events that dominate the epidemiology of SARS-CoV-2, yet the major unanswered question is whether natural selection played any role in the lineage distribution or if this was due to epidemiological factors such as fluctuation in lineage frequencies due to the random transmission events, i.e., founder effects[45].

Several genomic and environmental factors might explain the lineage turnover in Finland during 2020. The mutation rate of RNA viruses is considerably high, resulting in highly polymorphic virus populations. While the majority of mutations in the viral

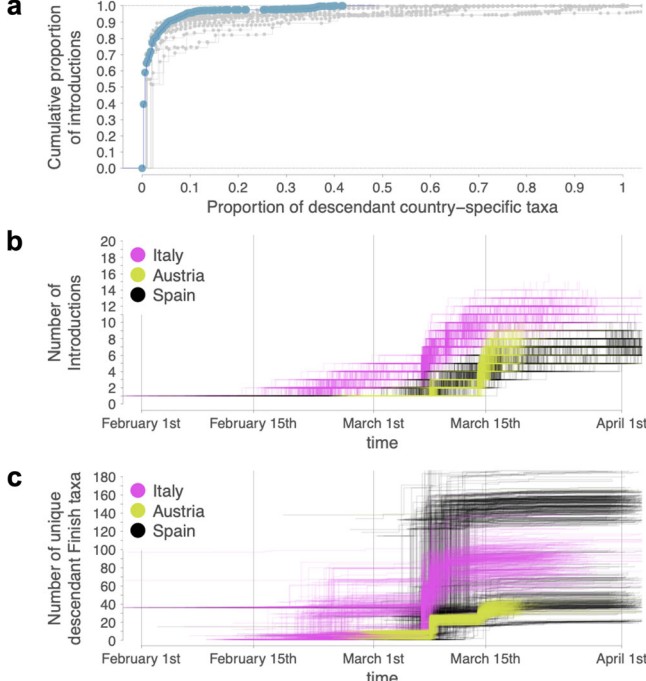

**Fig. 4 Empirical cumulative distribution function plot and temporal cumulative number of introductions. a** The proportion of descendant country-specific taxa (blue for Finland and gray for each of the 16 selected European countries excluding Latvia and Estonia for which few genomes were available) reflects the proportion of taxa from a given country that are descendants from a given introduction. The figure shows that the majority of introductions are responsible for a relatively small fraction of the taxa sampled in a given country, while a few introductions are responsible for a large proportion of the taxa sampled in a given country. Cumulative number of phylogeographic transitions (**b**) and cumulative number of unique Finnish descendant state taxa (**c**) over time from Italy, Austria and Spain to Finland.

genome are either neutral or lead to viral attenuation, mutations can occasionally result in higher fitness, such as more efficient transmission[46–48]. However, it is uncertain whether lineage turnover in Finland during 2020 was due to the higher fitness of introduced lineages. A more likely hypothesis would be that viral strains introduced to a new region with an immunologically naive population and relatively low incidence of infections become dominant due to epidemiological factors. This is exemplified by the high heterogeneity in the frequency of onward transmission of imported viruses with limited genetic diversity, as well as rapid turnover of circulating viruses during August and September in 2020. Regarding the latter, while there is some evidence that spike mutation D936Y may be positively selected, the dominant sub-cluster during late spring and summer containing this mutation was completely replaced by other lineages during this period. This is likely due to the low incidence of infections during late spring and summer. In such circumstances any lineage (B.1.36.22 and B.1.463 in this case) may become dominant due to the super-spreading events or other epidemiological factors. However, the potential biological factors affecting lineage turnover require further empirical investigation.

Using Bayesian phylogeographic analyses, we identified a total of 42 individual introductions in the ancestry of a sample of 333 genomes from Finland. This estimate of the relative contribution of external introductions in establishing local transmission chains is similar to the one observed in New York State (116 introductions in 828 sampled genomes)[49] but lower compared to Belgium

(331 introductions in 740 genomes)[50]. We show that the cumulative number of viral introductions from other European countries reached a plateau soon after border closure on the 19th of March (Fig. 3b), highlighting the impact of non-pharmaceutical interventions on containing viral spread.

Our analyses also indicate three countries as major sources of introductions, which are Austria, Italy and Spain. There were at least twice the number of returning travelers from Austria compared to our estimated viral introductions (Fig. 3b). One possible explanation for this observation is that travelers from Austria may have picked up the same source of infections in popular skiing resorts, resulting in their viral genome clustering (Supplementary Fig. 4). At least one ski resort known as Ischgl in Tyrol, Austria was recorded having an outbreak of SARS-CoV-2 in early March of 2020[51], which aligns with the Bayesian estimations. Furthermore, viral strains from this resort have been similarly detected and might have seeded transmission chains in several other European countries[52–54]. Italy being identified as another major source is notable, as the second imported case of SARS-CoV-2 to Finland was by a returning traveller from Milan, Italy. Both Italy and Spain might have contributed to the spread of SARS-CoV-2 in Finland due to them being popular tourist destinations. These countries also had their first outbreaks in February 2020[55,56], which aligns with our results and these infections have been linked to introductions of the virus in other countries[57–60] during early 2020. Our results suggest that if travel restrictions, quarantines, test-trace-isolation schemes or other border control forms are deployed in a timely manner, they may delay introductions developing to sustained community transmission. However, we would like to emphasize that these policies are likely to be effective only if they are combined with other preventive measures and if a given virus lineage or variant to be prevented has considerable and demonstrable incidence gradient over the border of the country of arrival.

To conclude, several genomic and epidemiological factors might have contributed to the rapid turnover of prevalent lineages among Finnish SARS-CoV-2 cases during the first wave in spring and the second one in fall of 2020. Our data suggest that the observed heterogeneity of detected virus cases is likely due to independent introductions from several neighboring and distant European countries, namely Austria, Italy and Spain, before imposing travel restrictions. In addition, we observed that the majority of circulating virus lineages were country-specific, mostly likely due to the high heterogeneity in the frequency of onward transmission of imported viruses.

### Data availability

All sequence data of this study (see accession IDs in Supplementary Data 2) are available in the GISAID database (https://www.gisaid.org/). Source data for Figs. 1, 3 and 4 are available in Supplementary Data 3.

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

## Acknowledgements

This study was supported by the Academy of Finland (grant number 336490), VEO—European Union's Horizon 2020 (grant number 874735), Finnish Institute for Health and Welfare (THL), the Jane and Aatos Erkko Foundation, and Helsinki University Hospital Funds (TYH2018322 and TYH2021343). We thank Kerstin Ahlskog for her technical assistance. FVdB was supported by the Research Foundation Flanders (Grant 1226120 N). PL and MAS acknowledge funding from the European Research Council under the European Union's Horizon 2020 research and innovation programme (grant agreement no. ~725422—ReservoirDOCS), from the Wellcome Trust through project 206298/Z/17/Z (Artic Network) and from the NIH grants R01 AI153044 and U19 AI135995. PL acknowledges funding from the EU grant 874850 MOOD. We acknowledge CSC-IT Center for Science, Finland for providing computational resources. Sequencing was performed by the Institute for Molecular Medicine Finland (FIMM) Genomics NGS Sequencing unit at University of Helsinki.

## Author contributions

Conceptualization: P.T.N., R.K., T.S.m., T.Si., O.V. Formal Analysis: P.T.N., R.K., T.S.m., F.V.dB., P.L. S.H. Funding acquisition: T.Si., O.V. Investigation: P.T.N., R.K., H.L., T.Sm. Methodology: P.T.N., R.K., S.H., P.E., T.Sm., F.V.dB., P.L., G.B., M.A.S. Project administration: R.K., T.Sm., O.V. Resources: P.T.N., R.K., F.V.dB., M.T.S., H.A., J.V., E.A., R.C., S.Ho., G.B., M.J.A., H.J., S.T.J., H.K.K., E.Ke., V.K., E.Ko., S.K., M.L., H.L., M.S., S.Ha., P.E., T.Si., P.L., O.V., T.Sm. Validation: P.T.N., R.K., H.L., S.K., M.L., P.E., T.Sm., T.Si., O.V. Writing—original draft: P.T.N., R.K., T.Sm., F.V.dB. Writing—review & editing: P.T.N., R.K., F.V.dB., M.T.S., H.A., J.V., E.A., R.C., S.Ho., G.B., M.J.A., H.J., S.T.J., H.K.K., E.Ke., V.K., E.Ko., S.K., M.L., H.L., M.S., S.Ha., P.E., T.Si., P.L., O.V., T.Sm.

## Competing interests

The authors declare that they have no competing interests.
