## [Peer Review File · Communications Medicine]

Reviewers' comments:

Reviewer #1 (Remarks to the Author):

The paper presents a phylodynamic analysis of the Finnish SARS-CoV-2 outbreak, using sequences from 2020. The authors start by break down the distribution of Nextstrain and Pagolin lineages over the course of the year. They then do a Bayesian phylogeographical reconstruction of imports into Finland during the first wave, using the Markov Jumps procedure, showing that most imports were from Italy, Spain and Austria. Finally, they look very briefly at the second wave in the autumn, with inconclusive results.

By-country molecular epidemiology analyses of the pandemic is a worthwhile project, and this has potential to be an important contribution to that. The overall approach and choice of dataset, at least for the first wave, is sensible. I very much appreciate the attention paid to sampling bias, both by thoughtful downsampling and by the sensitivity analyses that were performed. The manuscript does, however, feel disjointed, presenting two separate analyses of the two waves with unnecessarily divergent methodological approaches. The text also feels like it should have gone through a couple more rounds of drafting before submission. Finally, BEAST was definitely misconfigured and that analysis should be rerun.

There are two major issues with the BEAST analysis. The first is that the exponential tree prior was used (L395). The assumption here that the effective population size is undergoing exponential growth, but these sequences are from the entire Finnish first wave, including after the peak. This prior is therefore not appropriate, and should be swapped out; the most straightforward choice the skygrid. The dating of coalescent events under a misspecified tree prior will be biased, affecting figures 6B and 6C.

Secondly, I am not sure I understand why the Google mobility data was used. Lemey et al used it to demonstrate that it is a significant predictor of international spread, in an analysis that also tried several other potential predictors (see Table 1 of that paper). Here it is simply included. We are not told if the analysis supports its inclusion. BEAST is quite capable of estimating transition rates between countries on its own, without a predictor matrix. I am not clear if the authors do not understand this, or if they decided to use the Google matrix because there were problems with the simpler approach. If it is the latter, then this should be spelt out.

The analysis of the first wave uses a sophisticated phylodynamic model, while that of the second is much more rudimentary and simply asks us to inspect a maximum likelihood tree (figure 7A) and a timetree of uncertain origin (figure 7B). This feels rather like an afterthought, and is a shame – why not analyse the second half with BEAST as well? Even as it is, figure 7 is not very informative. We are told (L280) that “many of these sequences form three major clusters representing lineages B.1.36.22, B.1.463 and B.1.160” but these lineages are not marked. Figure S1 is a much more polished and informative diagram than this. The text ends with an extremely weak “However, the origin of these clades remained unresolved”. This statement is not really justified and certainly the authors show no signs of having tried to resolve those origins. Either get rid of the autumn section or do it properly.

On a final point, the paper does not really explore the interaction between the timeline of NPIs in Finland and the events they reconstruct from the genomics. For example, what was the effect of

border closures on the number of imports?

Minor points:

L58: SARS-CoV-2 is considerably more infectious than either SARS-CoV or MERS-CoV; the statement on infectivity is not correct.

Fig 1: Could the timeline of real-world events not be brought up to date with the analysis, i.e. to the end of 2020? When were international borders closed?

L116: What is meant “by the set national epidemic threshold”?

L143: The “respectively” seems misplaced here. You list four GISAID and Nextstrain lineages, but three Pangolin.

L151: A citation is needed for the statement re Sweden.

L168: I appreciate that there may be reasons for switching Nextstrain to Pangolin here, but please justify this in a bit more detail.

Fig 4: Use fewer colours; group unimportant lineages together.

L201: Is this a statement about the composition of the lineages making up the introductions in the first week of March, or of the overall distribution?

L205: How are there no identified diffusions from these countries? The mean points on figure 5B appear to be above zero. Is it just that they round to zero, or that the HPD includes zero?

L245: Is this the topological posterior probability of that clade, or the posterior probability for the country state of its root node?

Figure 6: In B and C the colours of the lines are barely discernible. I certainly cannot tell Italy from Spain.

L319: Is this pattern unusual for a country?

A general point on the methods section: much of the text is simply copy-and-pasted from the Lemey et al. preprint (<http://www.researchsquare.com/article/rs-208849/v1>). As Dr Lemey is an author on this work this is no major ethical failing; nevertheless, it is hardly good practice. In fact, there seems to be an error in the methods here that is derived from the other paper – the authors state that they “selected only sequences from the B.1 lineage with the D614G mutation for the analyses”. This text also appears in Lemey et al. (line 350) but in the current manuscript it does not appear even to be correct, as lineages 19A and 19B are included (figure 3) and these clades lack that mutation.

L347: The text states that the size of the dataset is 1,597, which covers both spring and autumn, but the text then charges off into talking about how the spring dataset was downsampled and augmented. At no point in this paper is the relative composition to the 1,597 of the spring and autumn waves given. It takes a lot of rereading to get to the point of realisation that this

augmentation procedure applied only to the spring sequences.

L365: presume that the autumn analysis used a different set of (746, from the legend of figure 7) international sequences, rather than a set sampled before May. But even if this is for the first wave, is not April a bit early to close the international dataset? The first wave was near its peak in May and ended in June. There could have been introductions later than April.

L369: But you discarded duplicates with identical sequences but different sampling dates? Is this justifiable? Wouldn't this whole step skew the distribution of descendants of each introduction?

L388: Are these two lines the entire methods section for the autumn phylogeny? No explanation is given of how these sequences was selected, nor of the IQ-TREE settings. I assume that the time tree comes from IQ-TREE's LSD procedure, not BEAST, but this is not stated either.

L408: The process by with the phylogeographic model was augmented with travel history information is entirely opaque. We are not given a reference to this procedure, if it refers to an analysis at all and not simply a comparison between that data and the phylogenetics results.

L425: It's not clear to me whether the results of the first "only sequence evolution" tree was just used as a starting tree for the phylogeography analysis, or whether it was fixed for the run.

Reviewer #2 (Remarks to the Author):

the manuscript is well written and of great interest

Reviewer #3 (Remarks to the Author):

Summary

This paper presents a rather basic analysis of the Finnish sarscov2 epidemic. It is written more like a report or thesis than a scientific paper. Far too much time and space are dedicated to extremely basic analyses such as subtyping using different methodologies, the results of which will not be interesting to anyone outside Finland. The phylogeographic analysis is well conducted and should form a larger part of the paper.

Major comments

- L139 – A lot more info is needed on sampling/sequencing strategy as it influences the results of phylodynamic analyses so much. Was it random? If a lot of samples were collected as a result of contact tracing, what downsampling method did you use to correct for that bias?
- In general, I think a lot of the descriptive analyses should be moved to the supplementary (e.g. lineages through time)
- The introduction is far too detailed about the history of the epidemic in Finland, and the genes/ORFs of sarscov2. Again this should be moved to supplementary. Instead, the introduction should talk about phylodynamic analyses, what they tell us, what they're for, why they can be useful, and give some examples from the literature.
- The figures are great – but there are too many of them and again, some should be moved to supplementary (Fig 2,3,4,7)

- The Results section contains too much Discussion/ Interpretation (e.g., comparison with other countries, description of effects of mutations)

Minor comments

It would be good to have a flow chart showing the number of sequences at each stage, this is not always clear. The abstract should indicate the number of sequences used in the main analysis.

L36 sarscov 2 WAS introduced to Finland

L38 – clade 20C will this mean anything to anyone?

L42 the source for a third of cases, unclear? Do you mean a third of cases were linked to an importation event?

L83 registry of the 13? Missing a word

Fig 1 is great!

Introduction: so much detail about number of cases through time in Finland but nothing about phylodynamics how it works and what it can do

L215 grammar

L233 42 or 43?

L252, I don't understand

L261 I am very pleased to see the sensitivity analysis with different sampling strategies for the phylogeographic analysis

L273 fig6A I don't understand what the proportion of descendant country-specific taxa is

Fig 7 – what does this tree show??

L347 – how were patients selected for sequencing? This is extremely important to the interpretation of your results.

L352 how did you choose these sequences from GISAID? Is it all sequences from the 16 countries up to April 17, 2020? This should be explained first

L357 what are google mobility flows?

L368 what is the logic behind eliminating duplicates

L390 IQTREE-2 (ref 35) is not a sarscov2 version of IQTREE

L425 I don't understand what you mean by this : “we initially only considered sequence evolution to arrive at a tree distribution from which trees were taken as starting trees in our phylogeographic analyses. “

Dear editor and reviewer(s),

Please find our point-by-point responses below in bold font. All text changes done in the revised manuscript and supplementary material are highlighted in yellow.

Reviewer #1:

The paper presents a phylodynamic analysis of the Finnish SARS-CoV-2 outbreak, using sequences from 2020. The authors start by break down the distribution of Nextstrain and Pagolin lineages over the course of the year. They then do a Bayesian phylogeographical reconstruction of imports into Finland during the first wave, using the Markov Jumps procedure, showing that most imports were from Italy, Spain and Austria. Finally, they look very briefly at the second wave in the autumn, with inconclusive results.

By-country molecular epidemiology analyses of the pandemic is a worthwhile project, and this has potential to be an important contribution to that. The overall approach and choice of dataset, at least for the first wave, is sensible. I very much appreciate the attention paid to sampling bias, both by thoughtful downsampling and by the sensitivity analyses that were performed. The manuscript does, however, feel disjointed, presenting two separate analyses of the two waves with unnecessarily divergent methodological approaches. The text also feels like it should have gone through a couple more rounds of drafting before submission. Finally, BEAST was definitely misconfigured and that analysis should be rerun.

We thank the reviewer for his/her critical appraisal. We agree that the manuscript may feel disjointed and we have therefore restructured the results into two parts: the first part provides a concise summary on lineage distribution and turnover during 2020, and the second part focuses on the Bayesian phylogenetic analyses of the first (spring) epidemic wave. The main reason for not performing a similar Bayesian analysis for the second wave epidemic is because we do not have individual travel information for sequences sampled after 17th of April 2020 (see below for more information). Regarding the configuration of BEAST, we believe there were several misconceptions about the data used for these analyses. Below, we rectify these misconceptions and provide additional sensitivity analyses to provide reassurance that our BEAST analysis was configured correctly.

There are two major issues with the BEAST analysis. The first is that the exponential tree prior was used (L395). The assumption here that the effective population size is undergoing exponential growth, but these sequences are from the entire Finnish first wave, including after the peak. This prior is therefore not appropriate, and should be swapped out; the most straightforward choice the skygrid. The dating of coalescent events under a misspecified tree prior will be biased, affecting figures 6B and 6C.

We believe there is a misunderstanding about the sequences used for the phylogeographic analyses. The exponential growth prior is appropriate for our BEAST analyses because we have only used viral sequences sampled up to the 17th of April (and not all first wave sequences as mentioned by the reviewer). This means that the large majority of sequences were sampled from a viral population experiencing exponential growth, and therefore an exponential tree prior seems appropriate. To demonstrate that our results are not sensitive to choice of the coalescent prior, we repeated the BEAST phylogeographic reconstruction with travel history using the skygrid as a tree prior. Our results show that the number and timing of introductions is similar as those obtained under a model assuming the exponential growth model (Supp. Fig. S6 and S7).

Secondly, I am not sure I understand why the Google mobility data was used. Lemey et al used it to demonstrate that it is a significant predictor of international spread, in an analysis that also tried several other potential predictors (see Table 1 of that paper). Here it is simply included. We are not told if the analysis supports its inclusion. BEAST is quite capable of estimating transition rates between countries on its own, without a predictor matrix. I am not clear if the authors do not understand this, or if they decided to use the Google matrix because there were problems with the simpler approach. If it is the latter, then this should be spelt out.

We agree that the information regarding the model support for mobility as a predictor of viral spread is lacking, and we apologize for this oversight. We estimated the inclusion probability of the Google mobility predictor as part of our inference. Our results confirm that Google mobility is a significant predictor of international viral spread, with an estimated posterior inclusion probability of 1 and a log coefficient of 0.5 (95% HPD interval = 0.36–0.65). This information is now added to the text for completion (lines 132–135).

As pointed out by the reviewer, recent work has already demonstrated that Google mobility is a strong predictor for international SARS-COV-2 spread (Lemey et al. 2021 Nature -

PMID: 34192736). Lemey and colleagues investigated viral spread between 10 European countries and considered international air transportation data, Google mobility data and the social connectedness index of Facebook as covariates of viral phylogeographical spread. The model offered strong support for mobility as a predictor of spatial diffusion in line with our results. This provides a strong motivation to include mobility as a predictor of international viral spread (here between 17 European countries) in our BEAST analyses to obtain well-informed phylodynamic estimates (instead of having to estimate $(17 \times 16) / 2 = 136$ or $17 \times 16 = 272$ in a standard symmetric or asymmetric model respectively).

The analysis of the first wave uses a sophisticated phylodynamic model, while that of the second is much more rudimentary and simply asks us to inspect a maximum likelihood tree (figure 7A) and a timetree of uncertain origin (figure 7B). This feels rather like an afterthought, and is a shame – why not analyse the second half with BEAST as well? Even as it is, figure 7 is not very informative. We are told (L280) that “many of these sequences form three major clusters representing lineages B.1.36.22, B.1.463 and B.1.160” but these lineages are not marked. Figure S1 is a much more polished and informative diagram than this. The text ends with an extremely weak “However, the origin of these clades remained unresolved”. This statement is not really justified and certainly the authors show no signs of having tried to resolve those origins. Either get rid of the autumn section or do it properly.

The main reason that we cannot repeat the Bayesian phylogeographic analyses for the second wave sequences is because we have no linked travel information for sequences sampled after the 17th of April, which is important for shaping the phylodynamic estimates as shown for the spring epidemic and as previously demonstrated by Lemey et al. 2020 (Nature Communications - PMID: 33037213). We agree with the reviewer that - compared to the analyses for the spring epidemic - the results for the autumn epidemic are less detailed. Hence, we propose a new structure for the results section whereby we i) provide a concise overview of the viral lineage distribution and turnover during the spring and fall epidemic waves in Finland, and ii) focus on the travel-aware phylogeographic analyses of the first-wave epidemic in Finland. We hope that our results are now more clearly presented.

On a final point, the paper does not really explore the interaction between the timeline of NPIs in Finland and the events they reconstruct from the genomics. For example, what was the effect of border closures on the number of imports?

We agree that we didn't explicitly mention this in the text, but we included this information in the inset of Figure 2B where we show that the cumulative number of introductions reached a plateau soon after border closure (19th of March 2020). We have now also added this information to the text (lines 265–267).

Minor points:

L58: SARS-CoV-2 is considerably more infectious than either SARS-CoV or MERS-CoV; the statement on infectivity is not correct.

The statement has been changed on lines 53–54.

Fig 1: Could the timeline of real-world events not be brought up to date with the analysis, i.e. to the end of 2020? When were international borders closed?

As there were no significant events related to SARS-CoV-2 in Finland after August, we did not extend the timeline to the end of 2020. The only notable events near the end of the year were recommendations e.g., distance working and university lectures in remote mode, which effects we could not predict.

L116: What is meant “by the set national epidemic threshold”?

The statement has been modified (now in supplementary text lines 39–40).

L143: The “respectively” seems misplaced here. You list four GISAID and Nextstrain lineages, but three Pangolin.

The sentence has been modified on lines 93–95.

L151: A citation is needed for the statement re Sweden.

We have added the appropriate citation (<https://doi.org/10.3390/v12091026>) on line 231.

L168: I appreciate that there may be reasons for switching Nextstrain to Pangolin here, but please justify this in a bit more detail.

Here we initially used both Nextstrain and pangolin nomenclature. As the genetic diversity of circulating SARS-CoV-2 sequences increased, the latter became more useful for more detailed analysis. This is due to pangolin classification labeling each individual viral lineage, while Nextstrain classifies the viruses into larger clades. We have now added the justification on lines 104–105.

Fig 4: Use fewer colours; group unimportant lineages together.

We have done the suggested changes to supplementary figure S2.

L201: Is this a statement about the composition of the lineages making up the introductions in the first week of March, or of the overall distribution?

This statement is about the overall distribution. We rephrased the sentence in the new version and hope that our statement is clear now (lines 136–137).

L205: How are there no identified diffusions from these countries? The mean points on figure 5B appear to be above zero. Is it just that they round to zero, or that the HPD includes zero?

The average number of estimated introductions from these countries is below 1 (ranging between 0.062 for Latvia and 0.653 for Turkey) and their HPD includes zero. We have added this information to the text for completion (lines 142–143).

L245: Is this the topological posterior probability of that clade, or the posterior probability for the country state of its root node?

The statement refers to the topological posterior probability of the clade.

Figure 6: In B and C the colours of the lines are barely discernible. I certainly cannot tell Italy from Spain.

We agree that the lines were insufficiently clear, and we have now increased the line widths in the figure (now figure 3).

L319: Is this pattern unusual for a country?

The described pattern is quite common for a country in 2020. When looking at the pattern in 2021, when the diversity of SARS-CoV-2 is higher, the pattern is even clearer.

A general point on the methods section: much of the text is simply copy-and-pasted from the Lemey et al. preprint (<http://www.researchsquare.com/article/rs-208849/v1>). As Dr Lemey is an author on this work this is no major ethical failing; nevertheless, it is hardly good practice. In fact, there seems to be an error in the methods here that is derived from the other paper – the authors state that they “selected only sequences from the B.1 lineage with the D614G mutation for the analyses”. This text also appears in Lemey et al. (line 350) but in the current manuscript it does not appear even to be correct, as lineages 19A and 19B are included (figure 3) and these clades lack that mutation.

We believe there is a misunderstanding about the sequences used for the phylogeographic analyses. It is true that Figure 3 in the previous version included all lineages, but for the BEAST analyses we have only selected sequences from the B.1 lineage with the D614G mutation — as correctly stated in our methods section.

Inevitably, some of the methods describe model specifications that have been used previously in studies employing Bayesian phylogenetic analyses of SARS-CoV-2. While this may come across as a copy-paste, we guarantee that our data and full model specification are unique to the study presented here.

L347: The text states that the size of the dataset is 1,597, which covers both spring and autumn, but the text then charges off into talking about how the spring dataset was downsampled and augmented. At no point in this paper is the relative composition to the 1,597 of the spring and autumn waves given. It takes a lot of rereading to get to the point of realisation that this augmentation procedure applied only to the spring sequences.

We agree that this is confusing, and we have restructured the methods section in two parts to match with the structure of the results section. We believe that this should improve the flow of the methods.

L365: presume that the autumn analysis used a different set of (746, from the legend of figure 7) international sequences, rather than a set sampled before May. But even if this is for the first wave, is not April a bit early to close the international dataset? The first wave was near its peak in May and ended in June. There could have been introductions later than April.

We agree that there may have been introductions later than April, although these will only represent a minor part of the total number introductions. Indeed, our results suggest that the cumulative number of introductions reached a plateau soon after border closure on the 19th of March (Figure 2B). As mentioned earlier, the main reason we only included sequences until the 17th of April is because we have no linked travel information for the Finnish sequences after this date. Hence, our Bayesian inference is dedicated to gain insights into the early stage of COVID-19 spread in Finland. This was clarified on lines 326–331.

L369: But you discarded duplicates with identical sequences but different sampling dates? Is this justifiable? Wouldn't this whole step skew the distribution of descendants of each introduction?

Identical sequences sharing the same location state (here country) are joined together into a single phylogenetic clade and will also share the same ancestral location state. Therefore, identical sequences with the same location state will not contribute to a better inference of the transition history, but they will increase computation time due to an increase in the number of taxa. Removing identical sequences is therefore a justifiable and commonly used approach for downsampling the number of sequences.

L388: Are these two lines the entire methods section for the autumn phylogeny? No explanation is given of how these sequences was selected, nor of the IQ-TREE settings. I assume that the time tree comes from IQ-TREE's LSD procedure, not BEAST, but this is not stated either.

We have now elaborated in further detail our sequence selection process and the IQ-TREE settings we used to compute the phylogenetic tree in the methods section (lines 293–316).

L408: The process by which the phylogeographic model was augmented with travel history information is entirely opaque. We are not given a reference to this procedure, if it refers to an analysis at all and not simply a comparison between that data and the phylogenetics results.

The sentence preceding line 408 (lines 370–372) provides an explanation and reference that motivate our decision to include travel history information in our analyses: "As the

ancestral reconstruction of locations depends on the availability of samples, over- or undersampling of sequences from a given location can greatly impact the estimated ancestral locations¹⁵. ". Reference 15 is the paper by Lemey et al. published in 2020 in Nature Communications titled "***Accommodating individual travel history and unsampled diversity in Bayesian phylogeographic inference of SARS-CoV-2***".

L425: *It's not clear to me whether the results of the first "only sequence evolution" tree was just used as a starting tree for the phylogeography analysis, or whether it was fixed for the run.*

This tree was used as a starting tree for the phylogeographic analyses. We have restructured this sentence to clarify our approach (lines 388–391).

Reviewer #2:

the manuscript is well written and of great interest

Thank you for the positive assessment of our manuscript.

Reviewer #3:

Summary

This paper presents a rather basic analysis of the Finnish sarscov2 epidemic. It is written more like a report or thesis than a scientific paper. Far too much time and space are dedicated to extremely basic analyses such as subtyping using different methodologies, the results of which will not be interesting to anyone outside Finland. The phylogeographic analysis is well conducted and should form a larger part of the paper.

We thank the reviewer for his/her suggestions, and we have significantly restructured the text that now provides a concise overview of the lineage distributions in 2020 while keeping the focus on the phylogeographic analyses. In addition, as suggested by the reviewer, we have moved parts of the introduction and some of the main Figures to supplementary material.

Major comments

- L139 – *A lot more info is needed on sampling/sequencing strategy as it influences the results of phylodynamic analyses so much. Was it random? If a lot of samples were collected as a result of contact tracing, what downsampling method did you use to correct for that bias?*

All data from 2020 was from HUSLAB. This contains samples from the hospitals and a couple from testing points from the border. No downsampling was used for the fall data. Data was random, however, the data might be biased for the most severe cases of SARS-CoV-2, and there was no contract tracing for our data at that time. This has now been added to the methods section (lines 301–303).

- *In general, I think a lot of the descriptive analyses should be moved to the supplementary (e.g. lineages through time)*

We agree with this and have thus moved descriptive sections of our main text to the supplementary material.

- *The introduction is far too detailed about the history of the epidemic in Finland, and the genes/ORFs of sarscov2. Again this should be moved to supplementary. Instead, the introduction should talk about phylodynamic analyses, what they tell us, what they're for, why they can be useful, and give some examples from the literature.*

We have moved the in-depth sections of the introduction to the supplementary section and modified the introduction as suggested.

- *The figures are great – but there are too many of them and again, some should be moved to supplementary (Fig 2,3,4,7)*

We agree with this comment and have formatted the manuscript accordingly.

- *The Results section contains too much Discussion/ Interpretation (e.g., comparison with other countries, description of effects of mutations)*

We agree and have formatted the section accordingly.

Minor comments

It would be good to have a flow chart showing the number of sequences at each stage, this is not always clear. The abstract should indicate the number of sequences used in the main analysis.

We have now added a flow chart (figure 4) visualizing the sequence data selection process from both spring and fall data sets.

L36 sarscov 2 WAS introduced to Finland

The sentence has now been corrected.

L38 – clade 20C will this mean anything to anyone?

The clade in mention is Finland/Sweden specific, thus making it important. The sublineage under 20C contains D936Y, which is consistent with the idea of super spreading from a limited number of introductions.

L42 the source for a third of cases, unclear? Do you mean a third of cases were linked to an importation event?

The sentence has been clarified (lines 41–42).

L83 registry of the 13? Missing a word

The sentence has now been corrected (line 78).

Fig 1 is great!

Thank you for the kind comment.

Introduction: so much detail about number of cases through time in Finland but nothing about phylodynamics how it works and what it can do

We have now added which research topics in our study can be investigated via phylodynamic studies (lines 70–71).

L215 grammar

The sentence has now been corrected (lines 151–152).

L233 42 or 43?

We apologize for this typo and have changed 43 to 42 (line 169).

L252, I don't understand

We believe the author is referring to the following sentence: "*To explore the sensitivity of our phylogeographic reconstructions to sampling bias, we incorporated unsampled taxa for 6 locations (Estonia, Latvia, Norway, Hungary, Poland and Turkey) that were represented by less than 60 sequences (other European countries were represented by at least 100 sequences)*". We are not entirely sure what caused confusion in this sentence. Perhaps the reviewer is not sure about the meaning of "unsampled taxa", which are sequences consisting of "N" nucleotides and associated with a location (here a given country with low sample size) and sampling times (for which we randomly sampled dates from case count distributions per country). We are happy to provide more information if needed.

L261 I am very pleased to see the sensitivity analysis with different sampling strategies for the phylogeographic analysis

We thank the author for his/her appreciation.

L273 fig6A I don't understand what the proportion of descendant country-specific taxa is

We agree this was not well explained. The proportion of descendant country-specific taxa reflects the proportion of taxa from a given country that are descendants from a given introduction. The figure shows that the majority of introductions (y-axis) are responsible for a relatively small fraction of the taxa sampled in a given country (x-axis), while a few introductions are responsible for a large proportion of the taxa sampled in a given country. This pattern is similar for all European countries and reflects the heterogeneity in SARS-CoV-2 transmission dynamics underlying the establishment of local transmission chains. We have added more information to the legend of figure 3 to clarify this.

Fig 7 – what does this tree show??

The main findings of this tree (Supplementary Fig. S3) were that most of the Finnish SARS-CoV-2 sequences in fall belonged to either lineage B.1.36.22, B.1.463 or B.1.160 based on phylogenetic analysis. Additionally, due to them each forming a monophyletic clade, it is probable that each was seeded by a single ancestor circulating in Finland during the fall of 2020. We have added this information to lines 115–121.

L347 – how were patients selected for sequencing? This is extremely important to the interpretation of your results.

All data from 2020 was from HUSLAB. This contains samples from the hospitals and a couple from testing points from the border. No downsampling for fall data. Patients were not selected based on any criteria i.e., sample collection was random. This has been clarified now on lines 301–303.

L352 how did you choose these sequences from GISAID? Is it all sequences from the 16 countries up to April 17, 2020? This should be explained first

Sequence selection procedure for spring data is already present in the text (lines 329–337).

L357 what are google mobility flows?

Google mobility flows represent de-identified aggregate flows of populations around the world computed from phones' location sensors. Hence, it is an estimate of the relative population flow between each pair of geographical areas over a given time interval. This information was present in the text (lines 326–329). For more detailed information on Google mobility flows, we would like to refer to the article by Kraemer et al. 2020 (Nature Human Behaviour - PMID 32424257).

L368 what is the logic behind eliminating duplicates

Please see our answer to a similar comment raised by the first reviewer (“L369: *But you discarded duplicates with identical sequences but different sampling dates? Is this justifiable? Wouldn't this whole step skew the distribution of descendants of each introduction?*”).

L390 IQTREE-2 (ref 35) is not a sarscov2 version of IQTREE

While the COVID-19 version of IQ-TREE2 is more optimized for running large data sets of SARS-CoV-2 sequences according to the developers, the underlying algorithm is fundamentally the same. Therefore, we think citing the original paper for IQ-TREE2 is appropriate and sufficient.

L425 I don't understand what you mean by this : "we initially only considered sequence evolution to arrive at a tree distribution from which trees were taken as starting trees in our phylogeographic analyses. "

Standard BEAST analyses usually initialize a Markov chain Monte Carlo analysis with a random starting tree. However, given the large dataset and complexity of the model in our paper, a good starting tree will help to speed up convergence because the tree topology is one of the most challenging parameters to estimate. Therefore, we start our analyses with a standard BEAST model considering only sequence evolution (strict molecular clock model, HKY nucleotide substitution model and exponential growth tree prior). The resulting tree is then used as a starting tree for our phylogeographic analyses. We have restructured this sentence to clarify our approach (lines 388–391).

Reviewers' comments:

Reviewer #1 (Remarks to the Author):

See attached file.

Reviewer #3 (Remarks to the Author):

I think the changes have hugely improved the manuscript, and I commend the authors' efforts. Given that travel bans are contentious and disputed, I would be interested in the authors including a few lines in their discussion on that point. What do the authors think these results mean for policy? To my mind, the fact that one introduction led to 1/3 of the Finnish epidemic highlights that even if immigration is kept to a minimum (e.g. essential travel), the downstream consequences can be dramatic, because most transmission happens within communities and not at the border. But people could use the paper to demonstrate the opposite if they were of the view that travel bans are a useful tool.

Manon Ragonnet

I thank the authors for the changes made. However, in my opinion a more considerable revision was needed than is presented here.

I continue to view the GLM approach as either poorly described or misguided. The Lemey et al preprint (DOI 10.21203/rs.3.rs-208849/v1) itself cites Lemey et al., 2014 (DOI 10.1371/journal.ppat.1003932) for methods. The formula for the predictor model can be found in Text S1 of that paper, at the bottom of page 5. In this analysis there is only a single flattened predictor matrix \mathbf{x}_1 , the Google mobility data. The only parameters to be estimated are β_1 , the multiplication factor of \mathbf{x}_1 , and δ_1 , an indicator variable taking values 0 and 1. When $\delta_1 = 0$, the entries of the rate matrix Λ are universally 1, which is an extremely simplistic model; as a result it is not very surprising that δ_1 is instead always equal to 1 in your MCMC chain. All that has been shown is a log-linear multiple of the mobility data matrix is superior to a model in which all transition rates are equal. That would be expected unless the Google mobility data was no better than uniform rates. That they are not is not really very convincing evidence of anything. Unless there is some undescribed modification to the method, I see no reason to use the GLM model with only one predictor. It gives BEAST very few options for rate matrices, when it is entirely possible to estimate the individual matrix entries instead.

I also stand by my concern that removing duplicates with different sampling will have effects, not on the inference of transition history, but, as I said in the initial review, the Finnish descendant count, which is explicitly presented in figure 3C. If this figure is intended to illustrate the proportion of Finnish cases with origins the three other countries, that downsampling step could significantly distort the picture. (A minor point, but I continue to be puzzled why a three-colour palette with two shades of pink was selected; this is unlikely to be colourblind-friendly.)

Clarity problems have not been addressed with sufficient care. I appreciate that the exponential growth coalescent model is appropriate if the only period analysed is the exponential phase; however that this is true is unclear not only in the initial submission but also *in this revision*, despite the attention this receives in the rebuttal document. Nowhere is it indicated that the 333 sequences were all sampled before 17 April. This cutoff is not even included as a step in the new flowchart. That the Lemey et al travel history model (DOI 10.1038/s41467-020-18877-9) was used is also extremely easy to miss in both versions. Furthermore, in at least one instance, text that the rebuttal reports as having been revised is actually unchanged (e.g. the text in parentheses in lines 141-143 of the original is identical to that in lines 93-95 of the revision, despite the statement in the rebuttal that “the sentence has been modified”).

Reviewer #1:

I thank the authors for the changes made. However, in my opinion a more considerable revision was needed than is presented here.

I continue to view the GLM approach as either poorly described or misguided. The Lemey et al preprint (DOI 10.21203/rs.3.rs-208849/v1) itself cites Lemey et al., 2014 (DOI 10.1371/journal.ppat.1003932) for methods. The formula for the predictor model can be found in Text S1 of that paper, at the bottom of page 5. In this analysis there is only a single flattened predictor matrix x_1 , the Google mobility data. The only parameters to be estimated are β_1 , the multiplication factor of x_1 , and δ_1 , an indicator variable taking values 0 and 1. When $\delta_1 = 0$, the entries of the rate matrix Λ are universally 1, which is an extremely simplistic model; as a result it is not very surprising that δ_1 is instead always equal to 1 in your MCMC chain. All that has been shown is a log-linear multiple of the mobility data matrix is superior to a model in which all transition rates are equal. That would be expected unless the Google mobility data was no better than uniform rates. That they are not is not really very convincing evidence of anything. Unless there is some undescribed modification to the method, I see no reason to use the GLM model with only one predictor. It gives BEAST very few options for rate matrices, when it is entirely possible to estimate the individual matrix entries instead.

We regret that we have not made the rationale for using the phylogeographic parameterization sufficiently clear in our revision. We would like to stress that we use the GLM with the Google mobility predictor in order to inform the phylogeographic reconstruction with this mobility data. This was also the main motivation in the recent work by some of authors (<https://www.nature.com/articles/s41586-021-03754-2>, for which the pre-print the Reviewer refers to was a predecessor), but as the Reviewer points out, in that work we were also able to evaluate the most appropriate mobility/connectivity predictor to inform our phylogeographic reconstructions. In the preprint reporting on that work, we were better able to spell out the motivation to inform such reconstructions with mobility (and other) data: "Phylogenetic reconstructions may be poorly resolved due to the relatively limited substitutions accumulating in SARS-CoV-2 over time. This is further confounded by the degree of mixing that can be expected from unrestricted travel prior to the lockdowns in spring 2020. Here, we perform a phylodynamic analysis to evaluate the relative importance of persistence versus new introductions in causing the resurgence in different European countries. To maximize the resolution of our reconstructions, we incorporate epidemiological data and measures of human mobility across countries." We

follow this motivation in the current work and also further build on the insight that Google mobility is the best mobility predictor to inform SARS-CoV-2 phylogeographic reconstructions. In addition to achieving better resolution, we also believe that this approach can offer protection against sampling biases, because discrete transition rates estimated from biased sampling may affect the ancestral reconstruction in an adverse way.

The Reviewer correctly interprets the parameterization of the GLM approach and we agree with the Reviewer that finding support for the mobility predictor over a model with uniform rates is a trivial test. This is exactly why we did not mention this in the first version of the manuscript, but added it in response to the Reviewer's previous comment that 'the predictor was simply included' and 'we are not told if the analysis supports its inclusion'.

We fully understand that BEAST is able to estimate the individual entries of the transition rate matrix, and some of our authors have also implemented a Bayesian stochastic search variable selection (BSSVS) procedure to make such inferences more efficient. However, even with the latter extension, this standard inference procedure is inferior to using the GLM parameterization as a convenient way to inform the reconstructions with mobility data (and estimate the size of its contribution) for the reasons mentioned above. However, it is possible that the Reviewer has a different opinion about this and therefore we now also sought for a more objective way of demonstrating why our GLM parameterization is a better option than estimating a high dimensional rate matrix. Specifically, we have now estimated marginal likelihoods for both using a path sampling (PS) and stepping stone sampling (SS) approach. To make this procedure efficient for the large data set investigated here, we average over the same set of empirical trees for both parameterizations. We reproduce the estimates below for the Reviewer's convenience, which demonstrate that the GLM model outperforms the standard model with BSSVS by over 200 ln marginal likelihood units:

Model	ln MLE, PS	ln MLE, SS
GLM	-1956.98	-1956.78
standard+BSSVS	-2186.41	-2186.39

The strong support for the GLM model is not surprising as many rate parameters need to be estimated from sparse data in the standard approach and even the BSSVS approach does not sufficiently improve its performance. To illustrate this, we plot below the posterior rate estimates for the standard model with BSSVS, which all remain relatively close to the prior distribution

(essentially an exponential distribution with a mean of 1). Of course, in the BSSVS approach, many of the rates will be associated with a zero rate indicator and hence sampled from their prior, but it remains challenging to identify a small subset of rates that are included in the model and considerably diverge from the prior.

In the revised manuscript, we now more explicitly motivate our choice for a phylogeographic approach informed with (Google) mobility data, also referring to the previous work on this, when mentioning the approach for the first time. We also do this in the methods section which now also includes the marginal likelihood procedure and the resulting estimates.

I also stand by my concern that removing duplicates with different sampling will have effects, not on the inference of transition history, but, as I said in the initial review, the Finnish descendant count, which is explicitly presented in figure 3C. If this figure is intended to illustrate the proportion of Finnish cases with origins the three other countries, that downsampling step could significantly distort the picture. (A minor point, but I continue to be puzzled why a three-colour palette with two shades of pink was selected; this is unlikely to be colourblind-friendly.)

We apologize for this oversight in the previous replies to the comments. We agree with the reviewer that downsampling duplicate sequences could potentially skew the results shown in Figure 3C. Therefore, we investigated the distribution and ancestry of duplicate sequences in the Maximum Clade Credibility tree. This revealed that there were 27, 28 and 15 additional Finnish descendants from introductions from Spain, Italy and Austria, respectively. This result indicates that the cumulative number of descendants as shown in Figure 3C will increase for each country, but it will not distort the overall pattern/conclusion of our study, namely that a Spanish introduction seeded considerably more Finnish descendants compared to Italian and Austrian introductions.

In the new version of the paper, we have i) corrected the y-axis label of Figure 3C from "Number of descendant Finnish taxa" to "Number of unique descendant Finnish taxa" to specify that these are observations obtained from analyses excluding duplicate sequences, and ii) included a sentence in the results section describing the numbers of additional Finnish (duplicate) descendants from the three countries to provide reassurance that the removal of duplicate sequences does not affect the overall pattern of Figure 3C (lines 193–199).

The three-colour palette in Figure 3B and C follows the country colour codes as displayed in Figure 1, which was done to maintain consistency. In the new version, we changed the colour for Spain to black.

Clarity problems have not been addressed with sufficient care. I appreciate that the exponential growth coalescent model is appropriate if the only period analysed is the exponential phase; however that this is true is unclear not only in the initial submission but also in this revision, despite the attention this receives in the rebuttal document.

In the revised manuscript, we added a few sentences to the methods section describing our motivation for using the exponential coalescent prior and the fact that we also repeated the BEAST phylogeographic reconstructions using the skygrid as a tree prior (lines 386–390). In the

last paragraph of the results section, it is described that our results are not sensitive to the choice of the coalescent prior. We hope that the reviewer now agrees that the choice of coalescent prior has sufficiently received attention in the revised manuscript.

Nowhere is it indicated that the 333 sequences were all sampled before 17 April. This cutoff is not even included as a step in the new flowchart. That the Lemey et al travel history model (DOI 10.1038/s41467-020-18877-9) was used is also extremely easy to miss in both versions.

We have now reworked the text in results where we first introduce the travel-aware phylogeographic inference to i) include information on the number of sequences and the fact that they were sampled until the 17th of April on lines 134–135 (this information was also added to the flowchart in Figure 4) and ii) to provide more information on the travel history model, including the reference to the Lemey et al. travel history paper (lines 136–140). We hope that we were able to better spell out our analyses in the revised version of the manuscript.

Furthermore, in at least one instance, text that the rebuttal reports as having been revised is actually unchanged (e.g. the text in parentheses in lines 141-143 of the original is identical to that in lines 93-95 of the revision, despite the statement in the rebuttal that “the sentence has been modified”).

We have modified this statement in this revision (lines 96–99) and apologize for this oversight on our part. We have additionally gone over the comments of the previous review to ensure that all points have been now addressed.

Reviewer #3:

I think the changes have hugely improved the manuscript, and I commend the authors' efforts.

Thank you for the kind comments.

Given that travel bans are contentious and disputed, I would be interested in the authors including a few lines in their discussion on that point. What do the authors think these results mean for policy? To my mind, the fact that one introduction led to 1/3 of the Finnish epidemic highlights that even if immigration is kept to a minimum (e.g. essential travel), the downstream consequences can be dramatic, because most transmission happens within communities at not at the border. But people could use the paper to demonstrate the opposite if they were of the view that travel bans are a useful tool.

The point presented by the reviewer here is important. The results are congruent with the known overdispersion in SARS-CoV-2 transmission and that a minority of cases is responsible for the majority of further infections (<https://doi.org/10.1073/pnas.2016623118>). Similarly, it is conceivable that few of the importations end up spreading in the society. Thereby, with the specific ability of controlling the importation of cases over international borders upon arrival, the likelihood of further downstream spreading can be in a larger scale limited proportional to importations prevented, but there may be large variability in the consequences of individual importations. We have now added this discussion in the manuscript as suggested (lines 301–306).

REVIEWERS' COMMENTS:

Reviewer #1 (Remarks to the Author):

My concerns have been addressed, thank you.

Reviewer #3 (Remarks to the Author):

I am satisfied with the authors' corrections and support publication.